# Role of Organic Cation Transporter 3 and Plasma Membrane Monoamine Transporter in the Rewarding Properties and Locomotor Sensitizing Effects of Amphetamine in Male andFemale Mice

**DOI:** 10.3390/ijms222413420

**Published:** 2021-12-14

**Authors:** Nikki J. Clauss, Wouter Koek, Lynette C. Daws

**Affiliations:** 1Department of Cellular and Integrative Physiology, University of Texas Health Science Center at San Antonio, San Antonio, TX 78229, USA; 2Department of Psychiatry and Behavioral Sciences, University of Texas Health Science Center at San Antonio, San Antonio, TX 78229, USA; koek@uthscsa.edu; 3Department of Pharmacology, University of Texas Health Science Center at San Antonio, San Antonio, TX 78229, USA

**Keywords:** organic cation transporter 3, plasma membrane monoamine transporter, amphetamine, conditioned place preference, locomotion

## Abstract

A lack of effective treatment and sex-based disparities in psychostimulant addiction and overdose warrant further investigation into mechanisms underlying the abuse-related effects of amphetamine-like stimulants. Uptake-2 transporters such as organic cation transporter 3 (OCT3) and plasma membrane monoamine transporter (PMAT), lesser studied potential targets for the actions of stimulant drugs, are known to play a role in monoaminergic neurotransmission. Our goal was to examine the roles of OCT3 and PMAT in mediating amphetamine (1 mg/kg)-induced conditioned place preference (CPP) and sensitization to its locomotor stimulant effects, in males and females, using pharmacological, decynium-22 (D22; 0.1 mg/kg, a blocker of OCT3 and PMAT) and genetic (constitutive OCT3 and PMAT knockout (−/−) mice) approaches. Our results show that OCT3 is necessary for the development of CPP to amphetamine in males, whereas in females, PMAT is necessary for the ability of D22 to prevent the development of CPP to amphetamine. Both OCT3 and PMAT appear to be important for development of sensitization to the locomotor stimulant effect of amphetamine in females, and PMAT in males. Taken together, these findings support an important, sex-dependent role of OCT3 and PMAT in the rewarding and locomotor stimulant effects of amphetamine.

## 1. Introduction

Amphetamine-like drugs are among the most commonly abused drugs worldwide, second only to cannabis [1]. Abuse of amphetamine-like psychostimulants is on the rise in North America, Oceania, and Asia [2,3,4]. Use of these drugs leads to a variety of adverse effects that place an undue burden on the public health system. In the short-term, these include, but are not limited to, restlessness, insomnia, hyperthermia, and convulsions. Long-term use can lead to addiction, paranoia, mood disturbances, agitation, psychosis, cognitive impairment, and death [5,6]. The impact of these effects is more likely to fall on individuals from demographics in which the use of amphetamine-like drugs is disproportionately popular, such as women [7]. For example, women begin psychostimulant use at younger ages, have increased acute responses, more rapidly escalate use, and progress to addiction faster than males [8,9,10]. Indeed, between 2016 and 2017, deaths from psychostimulant overdose increased by 33% and impacted more women than men [11].

A better understanding of the mechanisms by which amphetamine-like psychostimulants produce their abuse-related effects in both males and females will likely help to develop effective treatments. It is well known that amphetamine is a substrate for the high-affinity dopamine (DA), norepinephrine (NE), and serotonin (5-HT) transporters (DAT, NET and SERT, respectively). As a substrate for these transporters, amphetamine both inhibits uptake and stimulates release of monoamines through disruption of vesicular storage, thereby increasing extracellular monoamine levels via reverse transport [12,13,14,15]. The acute euphoric and rewarding effects of amphetamine are thought to be mediated primarily via DAT and the DA neurotransmitter system [14,16]. However, elegant work from the Jones’ group brought the importance of DAT in the actions of amphetamine into question, when they showed that amphetamine was able to increase extracellular DA in mice lacking DAT [17]. Moreover, DAT knockout (KO, −/−) mice developed robust conditioned place preference (CPP, a procedure which is often used to model abuse-related drug effects) for amphetamine, which persisted longer than CPP for amphetamine in wild-type mice [17]. Together with the failure of strategies targeting DAT, or SERT and NET, to treat amphetamine addiction [18,19], these findings suggest that amphetamine, and potentially related stimulants, have significant actions elsewhere to modulate dopaminergic neurotransmission.

A rapidly growing literature supports a prominent role for organic cation transporter 3 (OCT3) in regulating dopaminergic neurotransmission [20,21,22,23,24,25,26,27,28]. OCT3 is a bidirectional transporter with a low affinity, but high capacity (uptake-2 transport) for the non-selective transport of monoamines [29]. It is highly expressed in striatum and nucleus accumbens [30,31]—key regions in the rewarding effects of amphetamine [32]. Indeed, there are correlations between gene variants of OCT3 and methamphetamine abuse [33]. Data from our lab support the idea that OCT3 has a significant role to play in the ability of amphetamine to regulate dopaminergic transmission. We showed that the non-selective OCT and plasma membrane monoamine transporter (PMAT) inhibitor, decynium-22 (D22), inhibits amphetamine-evoked hyperlocomotion and DA release in vivo, effects that were lost in male constitutive OCT3−/− mice [24]. These data raise the possibility that OCT3 may be a novel target for the development of treatments for addiction to amphetamine and related psychostimulants.

In addition to OCT3, PMAT is another uptake-2 transporter that regulates dopaminergic neurotransmission [34,35,36]. In fact, PMAT transports DA more avidly than OCT3. Like OCT3, it is a bidirectional transporter, richly expressed in striatum and nucleus accumbens [31,34,36]. However, the role that PMAT might play in the actions of amphetamine has not been investigated.

Importantly, the roles of OCT3 and PMAT in the reinforcing effects of amphetamine, if any, remain unknown. Moreover, although females are more sensitive to the effects of psychostimulants and more vulnerable to their abuse, the role of these uptake-2 transporters in the actions of amphetamine is yet to be explored in females. Here, we begin to fill these knowledge gaps by investigating the roles of OCT3 and PMAT in the rewarding properties of amphetamine in both males and females. Because the conditioned place preference procedure used here to examine these rewarding properties also allowed concurrent assessment of amphetamine-induced locomotion and its sensitization, these latter effects were analyzed as well.

## 2. Results

### 2.1. CPP

#### 2.1.1. OCT3

Results support the hypothesis that D22 attenuates amphetamine induced CPP in OCT3+/+ male and female mice (Figure 1A; F (1,120) = 15.43, *p* = 0.0001), and does so independently of sex (pre-treatment × sex interaction: F (1,120) =0.39, *p* = 0.53). Our assessment of the hypothesis that D22 fails to attenuate amphetamine-induced CPP in male and female OCT3−/− mice showed that D22 does attenuate amphetamine-induced CPP in OCT3−/− mice, but does so in a sex-dependent manner, where D22 is only effective in preventing CPP for amphetamine in females (Figure 1B; D22 × sex interaction: F (1,120) = 4.40, *p* = 0.04; D22 in females: *p* = 0.002; D22 in males: *p* = 0.44). After saline pretreatment, amphetamine-induced CPP appeared to be less in OCT3−/− mice than in OCT3+/+ mice (F (1,120) = 3.45, *p* = 0.07) in both sexes (genotype × sex interaction: F (1,120) = 0.02, *p* = 0.89), with male OCT3−/− mice failing to develop statistically significant CPP for amphetamine.

CPP results with D22 + amphetamine in OCT3−/− females (Figure 1B, lower filled pink bar) suggested possible place aversion, because they showed a negative mean preference with 95% confidence limits that did not include zero (a value expected based on previously obtained data in saline-trained controls, Koek, 2016 [37]). To examine possible aversive effects of D22 when given alone, we conducted a CPP experiment in which males and females of both genotypes (*n* = 5–7 per group) received D22 as a pretreatment and saline as treatment. The analysis did not show statistically significant main- or interaction effects of sex and genotype (F (1,20) = 1.92, *p* = 0.18); the overall mean CPP, based on data obtained in 24 animals, was 160 and its 95% confidence limits (i.e., –110 to 420) included zero (data not shown).

#### 2.1.2. PMAT

Results support the hypothesis that D22 attenuates amphetamine induced CPP in PMAT+/+ male (*p* = 0.02; Figure 2A, blue bars) and PMAT+/+ female (*p =* 0.0002; Figure 2 left panel, pink bars) mice (pre-treatment × genotype × sex interaction: F (1, 20) = 6.28, *p* = 0.014). Our assessment of the hypothesis that D22 fails to attenuate amphetamine-induced CPP in female PMAT−/− mice but persists among male PMAT−/− mice was also supported (Figure 2B). Indeed, like PMAT+/+ mice, PMAT−/− males treated with D22 displayed significant attenuation of CPP (*p =* 0.004; Figure 2B, blue bars), whereas D22 did not attenuate CPP for amphetamine in female PMAT−/− mice (*p =* 0.03; Figure 2B, right panel, pink bars).

### 2.2. Locomotor Activity and Sensitization

#### 2.2.1. OCT3

There was a significant main effect of sex on locomotion during the first saline conditioning session, where, regardless of genotype, females made more beam breaks (mean = 958, SEM = 27) than males (mean = 757, SEM = 21) (F (1, 124) = 18.08, *p* < 0.0001) (Table 1). Because of these differences in basal locomotion, drug effects on locomotion during amphetamine conditioning sessions were expressed for each animal as a percentage of locomotion during the saline conditioning session that was conducted on the immediately adjacent day (for additional details of this approach, see [37]).

A four-factor ANOVA assessing the interaction of amphetamine-paired training day, pre-treatment, sex, and genotype was conducted and revealed a statistically significant interaction (F (2.8, 334.8) = 2.78, *p* = 0.045, Figure 3). 

To examine this four-way interaction in more detail, the interaction between amphetamine-paired training day and pre-treatment was assessed separately for each sex and genotype. In males, the interaction of amphetamine-paired training day and pretreatment was not significant in either OCT3+/+ (F (1.7, 51.8) = 1.96, *p* = 0.16) or OCT3−/− (F (2.6, 78.4) = 0.25, *p* = 0.84) mice. However, the interaction between amphetamine-paired training day and pre-treatment in OCT3+/+ females (F (2.1, 63,3) = 3.73, *p* = 0.027) suggests that the effect of amphetamine-paired training day is dependent on pre-treatment, such that a significant linear trend of locomotion across amphetamine-paired training days occurs in saline-pretreated (linear trend: *p* = 0.0008), but not in D22-pretreated (*p* = 0.16) OCT3+/+ females. Additionally, the interaction between amphetamine-paired training day and genotype in saline-pretreated females (F (1.6, 48.1) = 2.49, *p* = 0.10) indicates that saline-pretreated OCT3−/− females show less sensitization than saline-pretreated OCT3+/+ females.

#### 2.2.2. PMAT

There was a significant interaction effect of sex and genotype on locomotion during the first saline training session (F (1, 124) = 5.36, *p* = 0.02). Females (Mean = 1199, SEM = 45) made more beam breaks than males (M = 870, SEM = 27), regardless of genotype (Table 1). Additionally, while there were no genotype differences in baseline locomotion for males, PMAT+/+ females engaged in significantly more locomotor activity than PMAT−/− females (see Table 1). Because of these differences in basal locomotion, drug effects on locomotion during amphetamine training sessions were expressed for each animal as a percentage of locomotion during the saline-training session that was conducted on the immediately adjacent day (for additional details of this approach, see [37]).

There was no statistically significant four-way interaction of amphetamine-paired training day, pre-treatment, sex, and genotype (F (2.5, 300.8) = 1.14, *p* = 0.33, Figure 4). However, to compare results in PMAT animals with those in OCT3 animals, we further assessed the potential interaction between amphetamine-paired training day and pretreatment separately for each sex and genotype. The results of this analysis showed that amphetamine-induced locomotor sensitization occurs after saline pretreatment in male and female PMAT+/+ mice. This sensitization appears to be decreased after pretreatment with D22 (Figure 4); however, the interaction of amphetamine-paired training day and pretreatment was not significant in males (F (2.2, 70) = 1.66, *p* = 0.19) or females (F (1.6, 47.6) = 1.67, *p* = 0.20). In addition, sensitization to the locomotor stimulating effects of amphetamine was significantly attenuated in saline-pretreated PMAT−/− mice compared with saline-pretreated PMAT+/+ for both sexes (males: F (2.4, 71.1) = 34.05, *p* = 0.045, females: F (1.9, 56.2) = 3.76, *p* = 0.032). D22 did not markedly impact sensitization in PMAT−/− mice. There was a significant interaction of amphetamine-paired training day and pretreatment for PMAT−/− males (F (2.2, 65.5) = 4.43, *p* = 0.013); however, the maximal increase in locomotion was similar irrespective of pretreatment. There was no significant interaction between amphetamine-paired training day and pretreatment in females (F (2.3, 67.7) = 2.58, *p* = 0.076). Taken together, these results in PMAT+/+ and PMAT−/− mice suggest that sensitization to the locomotor effects of amphetamine is affected by PMAT, less so by D22, and similarly in males and females.

## 3. Discussion

Using a combination of pharmacological and genetic techniques, we provide evidence that OCT3 and PMAT are sex-dependently involved in the rewarding properties of 1 mg/kg amphetamine. To our knowledge, this is the first assessment of a role for OCT3 in the rewarding properties of amphetamine in both males and females, as well as the first assessment of the role of PMAT in the rewarding properties of amphetamine. As expected, we found that CPP for amphetamine develops in male and female OCT3+/+ and PMAT+/+ mice. The non-selective OCT/PMAT blocker, D22, robustly blocked CPP for amphetamine in all OCT3+/+ and PMAT+/+ mice, regardless of sex. This exciting finding supports the contention that DAT is not a major player in the rewarding effects of amphetamine and supports reports that CPP for amphetamine persists in DAT knockout mice [17]. D22 does not act at DAT, NET, or SERT [38], thus data from wild-type mice reveal one, or more, D22-sensitive transporters (OCT1-3 and PMAT) as critical for the rewarding effect of amphetamine, at least in terms of CPP. We have also previously established that in males, the effect of D22 to suppress amphetamine-induced locomotion is OCT3-dependent [24]. Unlike OCT1 and OCT2, PMAT has a similar profile to OCT3 in that it is widely expressed in brain, including regions important for reward, and avidly transports DA [39]. Thus, OCT3 and/or PMAT are likely players in the ability of D22 to suppress CPP for amphetamine in wild-type mice.

Consistent with a role for OCT3 in the rewarding properties of amphetamine, CPP for amphetamine did not develop in male OCT3−/− mice, and there were no detectable effects of D22. In contrast, CPP for amphetamine persisted in saline-pretreated female OCT3−/− mice, although it was modestly attenuated compared to female OCT3+/+ mice. Moreover, D22 not only blocked CPP for amphetamine in female OCT3−/− mice but produced place aversion. Our data show that D22 pretreatment followed by saline injection did not produce place preference or aversion in male or female mice of either OCT3 genotype, indicating that this dose of D22 likely does not lead to place preference or aversion by itself. These data raise the possibility that the combination of D22 and amphetamine results in an aversive reaction in OCT3−/− females. Further, that D22 pretreatment prevented CPP for amphetamine developing in female OCT3−/− mice suggests that D22 can suppress amphetamine induced CPP through mechanisms other than OCT3, such as the D22 sensitive uptake-2 transporter, PMAT.

We have previously established that in males, the effect of D22 to suppress amphetamine-induced locomotion is unlikely to be PMAT dependent [24]. The present findings suggest that in males, this also extends to the rewarding properties of amphetamine. However, in females, it appears that the rewarding properties of amphetamine are largely OCT3-independent. Thus, we investigated the role of PMAT in the rewarding properties of amphetamine. In contrast to our OCT3−/− cohort, saline-pretreated PMAT−/− males did develop CPP for amphetamine, which was blocked by D22-pretreatment. These data suggest that OCT3, and not PMAT, is crucial for the ability of D22 to prevent CPP for amphetamine in male mice. Consistent with our findings in OCT3−/− female mice, saline-pretreated PMAT−/− females developed CPP for amphetamine. However, in contrast to OCT3−/− female mice, D22 pre-treatment did not prevent CPP for amphetamine developing, and did not produce place aversion, suggesting that PMAT is necessary for place aversion to the combination of D22 and amphetamine in females. Together, these data suggest that PMAT, and not OCT3, is the primary driver of D22′s ability to attenuate CPP for amphetamine in female mice.

Consistent with our previous findings in male OCT3 mice [24], there was no difference in locomotor activity between male OCT3+/+ and OCT3−/− mice following a saline injection. This result is also consistent with studies, examining male animals only, showing no difference in basal locomotor activity between OCT3 genotypes [40,41], or PMAT genotypes [42]. Further consistent with our previous findings in OCT3 male mice [24], the ability of a single amphetamine (1 mg/kg) injection to stimulate locomotor activity was modest in males, regardless of genotype. In male PMAT−/− mice the ability of a single amphetamine injection to stimulate locomotion was dampened compared to their PMAT+/+ counterparts. As expected, based on our studies and those of others [42,43,44,45], female mice were generally more active than males following a saline injection, regardless of genotype. In addition, consistent with abundant literature showing greater sensitivity of females to the effects of psychostimulants [10,46,47,48,49,50,51], an acute amphetamine injection robustly increased locomotor activity in female mice, regardless of genotype.

In the assessment of sensitization to the locomotor stimulant effects of repeated amphetamine (1 mg/kg) administration among the OCT3 cohort, we found that female mice developed sensitization to the locomotor stimulating effects of amphetamine regardless of genotype. Although female OCT3−/− mice developed sensitization to amphetamine, their overall locomotor response to repeated amphetamine was greatly attenuated compared to their OCT3+/+ counterpart, implicating OCT3 in the locomotor stimulant effect of amphetamine. In addition, D22 attenuated sensitization in OCT3+/+ females, but had no effect in OCT3−/− females, consistent with our previous work in male mice [24]. Neither OCT3+/+ or OCT3−/− males developed sensitization to the locomotor stimulant effects of amphetamine regardless of saline or D22 pretreatment. These data suggest that in contrast to CPP for amphetamine, where OCT3 appears to play a major role in males, and not in females, OCT3 appears to play a role in development of sensitization to amphetamine in females, but not males. Of course, it must be recognized that only one dose of amphetamine was studied, and firm conclusions await future dose–response analyses.

Similar to OCT3 mice, PMAT females developed sensitization to the locomotor stimulant effects of amphetamine regardless of genotype. Likewise, although female PMAT−/− mice developed sensitization to amphetamine, their overall locomotor response to repeated amphetamine was attenuated compared to their PMAT+/+ counterpart, implicating PMAT in the locomotor stimulant effect of amphetamine. D22 pretreatment trended to attenuate sensitization to the locomotor stimulant effect of amphetamine in both female PMAT genotypes; however, the effects were not significant. Male PMAT mice followed similar trends. Sensitization to the locomotor stimulating effects of amphetamine occurred in both PMAT+/+ and PMAT−/− males; however, amphetamine stimulated locomotion in PMAT−/− males was less than PMAT+/+ counterparts, implicating a role for PMAT in the locomotor stimulant effect of amphetamine in males. D22-pretreatment trended to attenuate sensitization in PMAT+/+ mice. Taken together, these results suggest that OCT3 and PMAT play a role in the development of sensitization to the locomotor stimulant effects of amphetamine in females, as does PMAT in males. Although trends were apparent in other wild-type cohorts, the only marked effect of D22 pretreatment was to attenuate sensitization to the locomotor stimulant effect of amphetamine in female OCT3+/+ mice, suggesting that at this dose of amphetamine (1 mg/kg), D22 generally has no major impact. However, given our published findings [24] showing that D22 dramatically attenuates the locomotor stimulant effect of an acute injection of 3.2 mg/kg amphetamine in OCT3+/+ male mice, future dose–response studies will be informative.

The failure to develop sensitization in the male OCT3 cohort, as well as the pattern of sensitization among the female OCT3 cohort of mice is somewhat surprising, given their development of CPP, and that many aspects of CPP and sensitization overlap [51]. However, when psychostimulant-induced behaviors are simultaneously monitored in the same animal, apparent distinctions have been observed [52,53,54], which raises the possibility that these behaviors are governed by different neural substrates [54]. Our data from the OCT3 cohort of mice are consistent with this idea. In addition to this, we administered a single, relatively low dose of amphetamine, which likely played a role in these results for both males and females. For example, Shen and colleagues [55] demonstrated that amphetamine induced CPP and sensitization are induced by different doses of amphetamine in rats, with CPP inducible at a lower dose than sensitization. Another aspect that could impact these results is that the OCT3 and PMAT cohorts were tested separately. Therefore, a detailed examination of possible differences in sensitization between OCT3+/+ and PMAT+/+ males awaits future studies that vary the dose of amphetamine to assess dose–response, and that include all genotypes in the same experiments.

Finally, while drug metabolism in OCT3 and PMAT mice has not been investigated, these strains are both bred on a C57BL/6 background and show no overt behavioral or physiological phenotypes, thus it is unlikely that differences in drug metabolism account for the effects reported here.

Taken together, these results support a sex-specific role for OCT3 and PMAT in the rewarding and locomotor stimulant properties of amphetamine, and are consistent with prior work indicating that the role of DAT in these effects may be overstated [17]. Importantly, the translational utility of these results would be enhanced by more detailed studies of human brain OCTs and PMAT. For example, while OCT3 and PMAT appear to have similar expression and function in humans, detailed localization of brain OCT3 and PMAT have only been performed in rodents (see [56], for review). While it is still too early for clinical trials, results described here provide preclinical evidence of the therapeutic potential of targeting OCT3 and PMAT for treatment of dependence on amphetamine or its congeners. Future studies investigating the role of sex hormones in sex differences reported here will be of great interest, since there is essentially nothing known about how or whether sex hormones influence the function of brain OCTs and PMAT. Overall, these data point to OCT3 and PMAT as novel mechanisms contributing to sex-dependent variation in the rewarding and locomotor stimulant effects of amphetamine-like stimulants. These data have important implications for uptake-2 transporters as sex-specific targets for therapeutic intervention in the treatment of amphetamine addiction and encourage further research into the roles that OCT3 and PMAT play in the actions of amphetamine-like stimulants.

## 4. Materials and Methods

### 4.1. Subjects

The present study examined adult male and female wild-type (+/+) and constitutive knockout (−/−) OCT3 and PMAT mice. OCT3−/− and PMAT−/− mice were originally developed by Zwart et al., 2001 [57], and Duan and Wang, 2013 [34], respectively. Age ranges did not differ as a function of sex or genotype (Median = 121.5 days, range = 60 days). All mice were bred on a C57BL/6 background and obtained from our in-house colonies at the University of Texas Health Science Center at San Antonio (UTHSCSA). Animals were housed with same-sex littermates (2–5 per cage) in a temperature-controlled (24°C) vivarium maintained on a 14/10-hr light/dark cycle (lights on at 0700, experiments conducted during the light period) in plastic cages (Dimensions: 19 cm × 29.2 cm × 12.7 cm with a wire lid. Importantly, mice were not provided with environmental enrichment. Environmental enrichment is known to impact CPP, drug self-administration, and DAT activity in rodents [58,59,60,61,62,63,64]. While it is currently unknown whether environmental enrichment affects OCT3 or PMAT activity, it is highly likely, given its impact on other monoamine transporters. Mice were provided with Sani-chips bedding (Harlan Teklad, Madison, WI, USA) and ad-libitum access to food (Rodent sterilizable diet, Harlan Teklad, Madison, WI, USA) and water. Animals were maintained and experiments were conducted in accordance with the Institutional Animal Care and Use Committee at UTHSCSA, and with the Guide for the Care and Use of Laboratory Animals (https://grants.nih.gov/grants/olaw/Guide-for-the-Care-and-use-of-laboratory-animals.pdf; accessed on 12 September 2021).

### 4.2. Drugs

D22 and d-amphetamine sulfate were obtained from Sigma-Aldrich Co. (St. Louis, MO, USA), dissolved in physiological saline, and injected intraperitoneally in a volume of 10 mL/kg. Doses are expressed as the weight of the salt. The D22 dosage of 0.1 mg/kg was used based on our previous work demonstrating that this dosage elicits behavioral effects [24,37,65,66] without impairing locomotor activity.

The amphetamine dosage used (1 mg/kg) was based on work demonstrating that doses between 1–3 mg/kg amphetamine (but not lower, e.g., [67]) induce CPP in male C57BL/6J mice [68], and in mice genetically modified on a C57BL/6J background [40,69] Therefore, 1 mg/kg may be the lowest dose of amphetamine with near maximal CPP-inducing effects in male C57BL/6J mice. If this is the case, CPP induced by 1 mg/kg amphetamine could afford a more sensitive measure to examine its hypothesized attenuation by D22 than CPP induced by higher doses.

### 4.3. Apparatus

Eight 30 cm × 15 cm × 15 cm customized acrylic boxes (Instrumentation Services, UTHSCSA, San Antonio, TX, USA), separately enclosed in commercially available, light, and sound-attenuating chambers (model no. ENV-022M, Med Associates, St. Albans, VT, USA), were used. Location in the chamber and locomotor activity were detected using six sets of infrared photodetectors (6 cm intervals, 2 cm above the floor) that were mounted along the sides of each conditioning box. Occlusion of the infrared light beams were counted using Multi-Varimex computer software (v2.10, Columbus Instruments, Columbus, OH, USA).

The floors of the conditioning boxes were metal, removable, and varied in texture (either a grid or a hold texture) across conditions. Grid floors were made up of 2.3 mm stainless steel rods that were mounted in parallel 6.4 mm apart. Hole floors were made from stainless steel sheets perforated with evenly distributed 6.4 mm round holes on 9.5 mm staggered centers. The floors to measure preference were half grid and half hole.

### 4.4. Conditioning Procedure

Conditioned Place Preference (CPP) in rodents, which is often used to model abuse-related drug effects, is shown by a preference for an environment previously paired with a drug compared with an environment previously paired with vehicle [70,71]. Conditioned place preference (CPP) and sensitization of locomotor activity were measured in the same animals using a procedure similar to that described previously [37]. All mice in the current experiments were exposed to the often used, unbiased, one-compartment place-conditioning procedure, with sessions conducted once per day [37,72]. To counterbalance tactile stimulus (floor type) and drug assignment (saline or D22 pretreatment), amphetamine was paired with the hole floor-texture and the vehicle with the grid floor-texture for half of the mice; pairings were opposite for the remaining mice. The CPP procedure consisted of three phases (Figure 5): habituation (one session), conditioning (eight sessions), and place preference test (one session). Immediately before each session, each animal received an intraperitoneal injection of saline or 0.1 mg/kg D22, followed 60 min later by a second injection of saline or 1 mg/kg amphetamine (n = 16 per treatment). After the second injection, mice were immediately placed in the center of the apparatus. Between tests, the floor and the inside of the boxes were wiped, and the litter paper beneath the floor was replaced. The habituation session was intended to reduce the novelty and stress associated with handling, injection, and exposure to the apparatus; thus, for this session, all mice received saline and were placed in the apparatus for 30 min on a floor covered with paper. The following 8 days, 30 min conditioning sessions were held. Conditioning consisted of pairing one floor type with the injection of amphetamine and the other floor type with the injection of saline. The day after the last conditioning session, the 30 min floor preference test was conducted in mice having received saline. The time spent on the amphetamine-paired floor was subtracted from the time spent on the saline-paired floor, and this difference was used to measure place preference. Previous reports [37,73] demonstrate that C57BL/6J control mice repeatedly treated with only saline spent the same amount of time on both floor types during preference tests, indicating no preference for one floor type over the other, and allowing for the use of an unbiased method to assess CPP. Sensitization to the locomotor stimulant effects of amphetamine, evidenced by progressively enhanced locomotor responses following repeated administration of amphetamine [51], was measured during the amphetamine conditioning sessions of the CPP procedure. All sessions were conducted once per day between the hours of 8 AM and 11 AM.

### 4.5. Data Analyses

All analyses were conducted with GraphPad Prism version 9.00 for macOS (GraphPad Software, La Jolla, CA, USA), except the 4-factor analysis of variance (ANOVA) of amphetamine-induced locomotion, which was analyzed with the R package afex (Analysis of Factorial Experiments) implemented in jamovi (www.jamovi.org). Statistical significance was defined as *p* < 0.05. We first assessed the hypotheses that 1) D22 attenuates amphetamine-induced CPP in OCT3+/+ male and female mice, and (2) D22 fails to do so in OCT3−/− male and female mice. Results of these analyses indicated the possibility that there may be a sex-specific role for other uptake-2 transporters, such as PMAT (see below). Therefore, we also assessed the hypotheses that 1) D22 attenuates amphetamine-induced CPP in PMAT+/+ male and female mice, and 2) D22 fails to do so in PMAT−/− females, but has no effect on PMAT−/− males. To assess CPP, data were analyzed by three-factor ANOVA, with pre-treatment (D22 vs. saline), genotype (OCT3+/+ vs. OCT3−/− or PMAT+/+ vs. PMAT−/−), and sex as independent variables, and CPP (time spent on the amphetamine paired floor-time spent on the saline paired floor) as the dependent variable. Three-way ANOVAs were followed by planned two-factor ANOVAs to further probe differences between sex and genotypes. Post hoc multiple comparison tests with Tukey’s correction were carried out to probe any significant interaction effects.

Based on CPP results in OCT3 animals pretreated with D22 or saline (see below), an additional CPP experiment was conducted to assess the impact of D22, without amphetamine, on place preference in OCT3+/+ and OCT3−/− mice. A two-factor ANOVA with genotype (OCT3+/+ vs. OCT3−/−) and sex as independent variables, and CPP (time spent on the D22-paired floor-time spent on the saline paired floor) as the dependent variable was used to analyze the data.

To assess sensitization to the locomotor-stimulant effects of amphetamine, measured by the increase in beam breaks during the 4 amphetamine conditioning sessions, locomotion during each amphetamine conditioning session was expressed for each animal as a percentage of locomotion during the corresponding (1^st^ to 4^th)^ saline conditioning session. Changes in locomotion during the conditioning phase were analyzed by four-factor ANOVA followed separately for each by genotype and sex by ANOVAs with pre-treatment (saline vs. D22) as between subject factor, amphetamine conditioning day as within-subjects factor, and trend analysis of the amphetamine conditioning day effects, to test the hypotheses that (1) D22 attenuates sensitization to the locomotor effects of amphetamine in OCT3+/+ and PMAT+/+ male and female mice, and (2) D22 fails to do so in OCT3−/− and PMAT−/− mice in a sex-dependent manner. Because of violations of the sphericity assumption, detected with Mauchly’s test, Greenhouse–Geisser corrected-repeated measures analyses of treatment effects on sensitization were conducted separately by genotype and sex. To assess effects of genotype, follow-up analyses were performed, consisting of separate ANOVAs with genotype as between subject factor and amphetamine conditioning day as within subject factor.

## Figures and Tables

**Figure 1 ijms-22-13420-f001:**
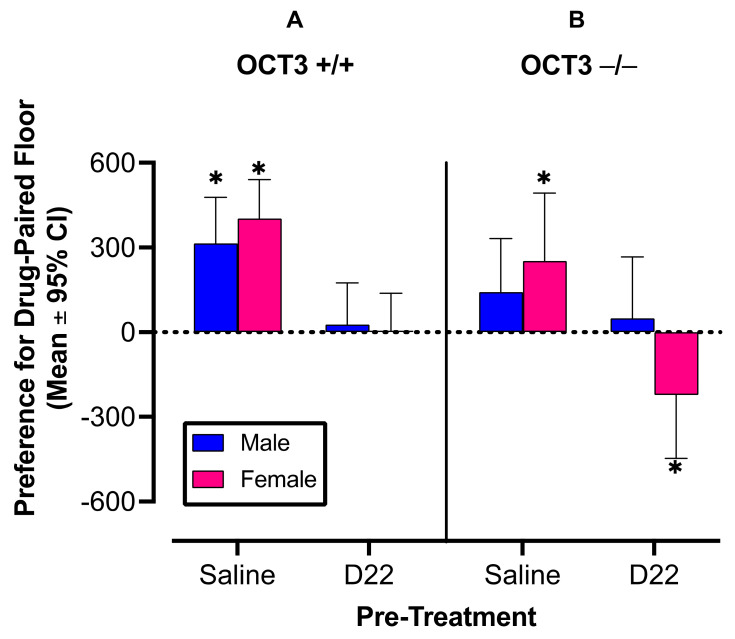
Decynium 22 (D22) prevents amphetamine-induced conditioned place preference (CPP) in organic cation transporter 3 (OCT3) wild-type (+/+) male and female mice (**A**), and in female OCT3 knockout (−/−) mice (**B**). Male OCT3−/− mice do not develop significant CPP for amphetamine. n = 16 per group. Error bars indicate 95% confidence intervals (CI). Asterisks indicate mean values with 95% CI that do not include zero, representing statistically significant place preference (male and female OCT3+/+ and female OCT3−/− mice) or aversion (female OCT3−/− mice). * *p* < 0.05

**Figure 2 ijms-22-13420-f002:**
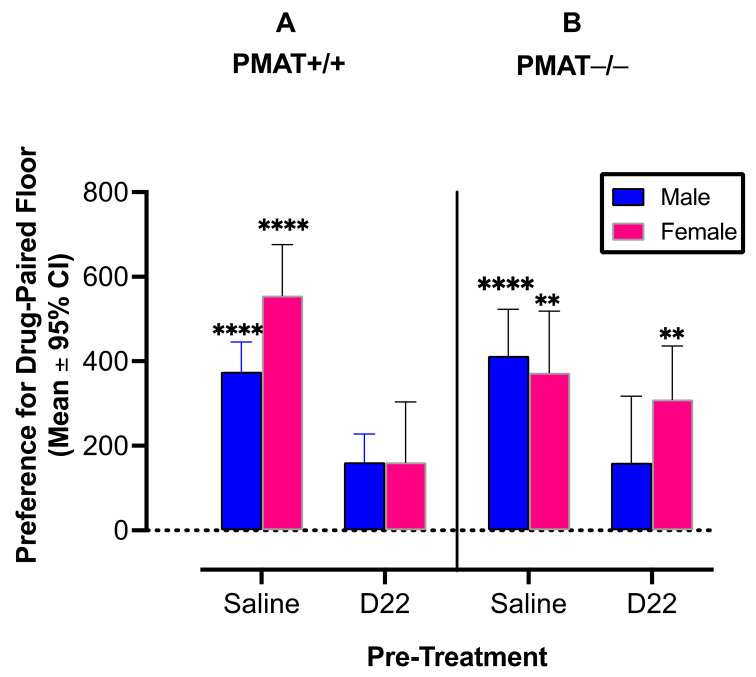
Decynium 22 (D22) prevents amphetamine-induced conditioned place preference (CPP) in plasma membrane monoamine transporter (PMAT) wild-type (+/+) male and female mice (**A**), and in male PMAT knockout (−/−) mice (**B**). Female PMAT−/− mice do not develop significant CPP for amphetamine. n = 16 per group. Error bars are 95% confidence intervals (CI). Asterisks indicate mean values with 95% CI that do not include zero, representing statistically significant place preference (male and female PMAT+/+ and female PMAT−/− mice). ** *p* < 0.01; **** *p* < 0.0001

**Figure 3 ijms-22-13420-f003:**
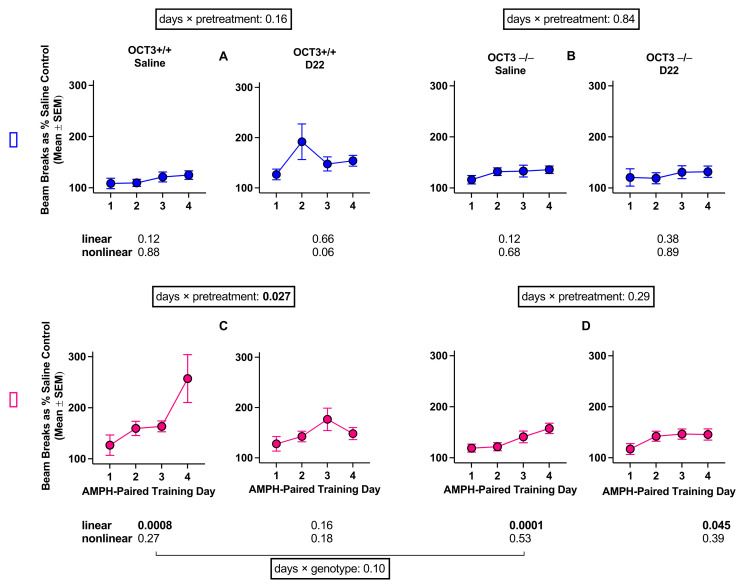
Sensitization to the locomotor stimulating effects of amphetamine (AMPH) in male (**A**,**B**) and female (**C**,**D**) organic cation transporter 3 (OCT3) wild-type (+/+) and OCT3 knockout (−/−) mice treated with saline or decynium 22 (D22) before amphetamine. Each part of the figure (**A**–**D**) shows the *p* value for the days by pretreatment interaction for each genotype and each sex. In addition, a *p* value is shown for the comparison of genotypes in saline-pretreated females. Additionally, each of the 8 panels (n = 16 mice per panel) shows *p* values for linear and nonlinear trends across days. Statistically significant *p* values are shown in bold.

**Figure 4 ijms-22-13420-f004:**
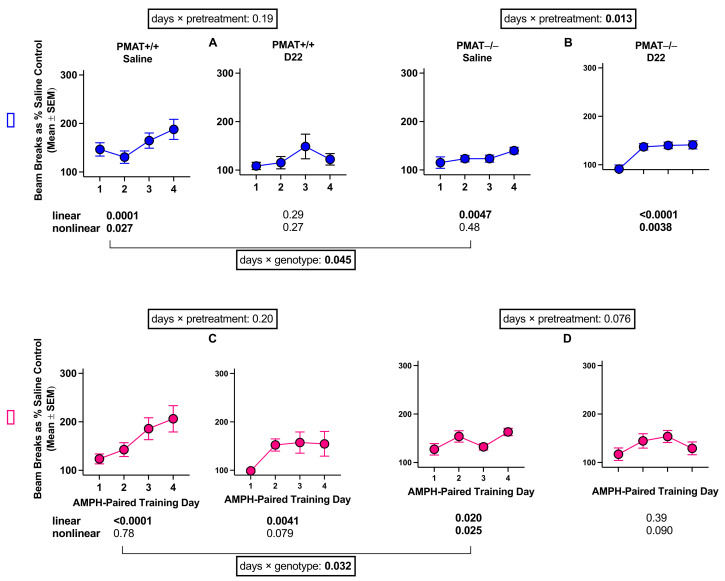
Sensitization to the locomotor stimulating effects of amphetamine (AMPH) in male (**A,B**) and female (**C,D**) PMAT+/+ and PMAT−/− mice treated with saline or decynium 22 (D22) before amphetamine. Each part of the figure (**A–D**) shows the *p* value for the days by pretreatment interaction for each genotype and each sex. In addition, a *p* value is shown for the comparison of genotypes in saline-pretreated males and females. Additionally, each of the 8 panels (n = 16 mice per panel) shows *p* values for linear and nonlinear trends across days. Statistically significant *p* values are shown in bold.

**Figure 5 ijms-22-13420-f005:**
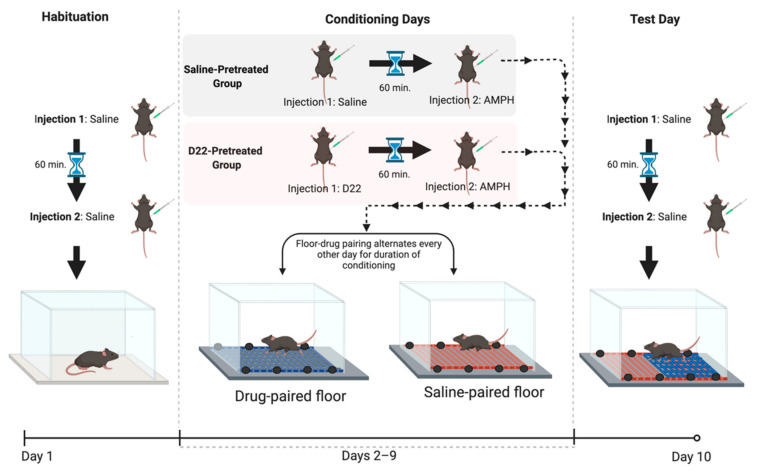
Schematic of conditioned place preference procedure (CPP). The CPP procedure consisted of three phases: habituation (one session), conditioning (eight sessions), and place preference test (one session). Immediately before each daily session, each animal received an intraperitoneal injection of saline or 0.1 mg/kg decynium 22 (D22), followed 60 min later by a second injection of saline or 1 mg/kg amphetamine (AMPH). After the second injection, mice were immediately placed in the center of the apparatus. For the habituation session, all mice received saline and were placed in the apparatus for 30 min (min) on a floor covered with paper. Thirty minute conditioning sessions were held on the following 8 days. Conditioning consisted of pairing one floor type with the injection of amphetamine and the other floor type with the injection of saline. The day after the last conditioning session, the 30 min floor preference test, using a floor in which each half consisted of each floor type, was conducted in mice having received saline. Place preference was measured by subtracting the amount of time spent on the drug-paired floor from the amount of time spent on the saline-paired floor. Figure created with Biorender.com.

**Table 1 ijms-22-13420-t001:** Mean beam breaks during the first saline conditioning session and the first amphetamine conditioning session in males and females in the saline and decynium 22 (D22) pretreatment groups. Numbers in parentheses are standard error of the mean (SEM).

Genotype	OCT3+/+		OCT3−/−		PMAT+/+		PMAT−/−	
Sex	Male	Female	Male	Female	Male	Female	Male	Female
**Saline**	768 (42)	944 (56)	747 (37)	973 (52)	821 (40)	1279 (65)	921 (33)	1119 (59)
**Saline + Amphetamine**	808 (75)	1055 (79)	859 (66)	1090 (98)	1015 (30)	1448 (63)	1084 (146)	1415 (97)
**D22 + Amphetamine**	903 (58)	1067 (51)	805 (75)	1119 (85)	929 (72)	1255 (75)	822 (80)	1186 (110)

## Data Availability

Data can be provided upon request to claussn@uthscsa.edu or daws@uthscsa.edu.

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
