# Peer review of "Role of Organic Cation Transporter 3 and Plasma Membrane Monoamine Transporter in the Rewarding Properties and Locomotor Sensitizing Effects of Amphetamine in Male andFemale Mice"

_ijms, 2021, doi:10.3390/ijms222413420_

Round 1
Reviewer 1 Report
Comments to the Authors
Manuscript ID: ijms-1467091
Title: Role of organic cation transporter 3 and plasma membrane monoamine transporter in the rewarding properties and locomotor sensitizing effects of amphetamine in male and female mice
Using a combination of pharmacological and genetic techniques, authors provided evidence that organic cation transporter 3 (OCT3) and plasma membrane monoamine transporter (PMAT) are sex-dependently involved in the rewarding properties of amphetamine. The obtained results, basing on statistical analyzes, revealed that OCT3 is necessary for development of conditioned place preference (CPP) to amphetamine in males, whereas in females, PMAT is neccesary for the abilty of decynium-22 (D22) to prevent development of CPP to amphetamine. Both OCT3 and PMAT appear to be important for development of sensitization to the locomotor stimulant effect of amphetamine in females, and PMAT in males.
The subject of manuscript is adequate to its content. The material and methods are described correctly. The results are presented in understandable way. A discussion based on available literature seems to be exhaustive. In my opinion, the manuscript may be ready for publishing, after the corrections recommended below.
- In the text, reference should be made to Scheme 1.
Material and methods
- The figure 5 mentioned in the text is missing (line 384).
- Line 460 - lack of bracket
- There was a problem opening the linked page (line 345). An information appeared on the website: To report a broken link, please E-mail OER Webmaster.
References
- The word doi is missing in the last 3 literature items (line 642-647).
- Please use a space to separate words or numbers in the literature.

Author Response
|
Intro |
In the text, reference should be made to Scheme 1. |
Scheme 1 was meant as a graphical abstract and was shifted without our noticing. We have moved it back to the abstract. |
|
Materials and methods |
The figure 5 mentioned in the text is missing (line 384). |
Figure 5 was somehow deleted from the document without our knowledge. It has been added back. |
|
|
Line 460 - lack of bracket |
Thank you for noticing this. We have added the end bracket. |
|
|
There was a problem opening the linked page (line 345). An information appeared on the website: To report a broken link, please E-mail OER Webmaster. |
We believe we have remedied this by re-applying the hyperlink. |
|
References |
The word doi is missing in the last 3 literature items (line 642-647). |
Thank you for noticing this detail. We have added the word doi to the items in question. |
|
|
Please use a space to separate words or numbers in the literature. |
This has been done. |
Reviewer 2 Report
I propose the following additions to the manuscript:
- Are the OCT3 and PMAT transporters the same as in humans? Do they have identical metabolism confirmed?
- How can the obtained results be translated into clinical practice?
- Didn't the mice differ significantly in body weight over such a wide age range?
- Did it not affect the biochemical parameters and then the obtained results?
- Whether animals were anesthetized during intraperitoneally administration?
- Did the animals have cage variety?
Author Response
|
Are the OCT3 and PMAT transporters the same as in humans? Do they have identical metabolism confirmed? |
OCT3 and PMAT appear to have similar expression and function in humans, though detailed localization of brain OCTs and PMAT have only been performed in rodents (see Koepsell (2021) for an overview). In terms of metabolism, to our knowledge, this has not been systematically investigated. Since both strains are bred on a C57BL/6 background and show no overt behavioral or physiological phenotypes, it is unlikely that differences in drug metabolism account for the effects reported here. We have added text to this effect in the revised manuscript. |
|
How can the obtained results be translated into clinical practice? |
While it is still too early for clinical trials, these results support previous preclinical results demonstrating the therapeutic potential of targeting OCT3 or PMAT for the treatment of dependence on amphetamine or its congeners. We have added text to this effect in the revised manuscript. |
|
Didn't the mice differ significantly in body weight over such a wide age range? |
Bodyweight did not differ significantly due to the mice being at least P90 at the time of treatment. |
|
Did it not affect the biochemical parameters and then the obtained results? |
All doses were administered based on each individual animals’ weight and therefore did not likely interfere with biochemical parameters or results. |
|
Whether animals were anesthetized during intraperitoneally administration? |
Animals were not anesthetized during i.p injections as this would likely interfere with the action of amphetamine and D22. Importantly, anesthesia could interfere with the behaviors being measured. |
|
Did the animals have cage variety? |
Whether environmental enrichment affects OCT3 activity is unknown, but highly likely given it impacts other monoamine transporters. Mice were therefore not provided with environmental enrichment. All animals were housed with same-sex littermates.
We have added these details to the discussion. |